# Numerical Simulation of the Advantages of the Figure-Eight Flapping Motion of an Insect on Aerodynamics under Low Reynolds Number Conditions

**DOI:** 10.3390/biomimetics9040249

**Published:** 2024-04-20

**Authors:** Masato Yoshida, Tomohiro Fukui

**Affiliations:** 1Department of Master’s Program of Mechanophysics, Kyoto Institute of Technology Matsugasaki Goshokaido-cho, Sakyo-ku, Kyoto 606-8585, Japan; m2623035@edu.kit.ac.jp; 2Department of Mechanical Engineering, Kyoto Institute of Technology Matsugasaki Goshokaido-cho, Sakyo-ku, Kyoto 606-8585, Japan

**Keywords:** unmanned aerial vehicles, hovering motion, elevation angle, Reynolds number, lift coefficient, power coefficient, vortex

## Abstract

In proceeding with the advanced development of small unmanned aerial vehicles (UAVs), which are small flying machines, understanding the flight of insects is important because UAVs that use flight are attracting attention. The figure-eight trajectory of the wing tips is often observed in the flight of insects. In this study, we investigated the more efficient figure-eight motion patterns in generating lift during the hovering motion and the relationship between figure-eight motion and Reynolds number. For this purpose, we compared the ratios of the cycle-averaged lift coefficient to the power coefficient generated from each motion by varying the elevation motion angle, which is the rotational motion that represents the figure-eight motion, and the Reynolds number. The result showed that the motion with a smaller initial phase of the elevation motion angle ($\varphi_{e0}\leq 90{}^{\circ}$) could generate lift more efficiently at all Reynolds numbers. In addition, the figure-eight motion was more effective when the Reynolds number was low.

## 1. Introduction

Research on small unmanned aerial vehicles (UAVs) is attracting attention in military and civilian applications [1]. There are three main types of small UAVs: micro air vehicles (MAVs), nano air vehicles (NAVs), and pico air vehicles (PAVs). The size of each UAV is defined as follows. For MAVs, the Defense Advanced Research Projects Agency (DARPA) defined UAVs as having a size within 15 cm, a mass of 50–100 g, a flight speed of 30–60 km/h, and a flight duration of 10 km as the goal of developing UAVs by Broad Agency Announcement (BAA 97-29) in 1997 [2]. In addition, for NAVs, the DARPA defined UAVs as having a size within 7.5 cm, a mass of 10 g, a flight speed of 18–36 km/h, and a flight duration of 1 km as the goal of developing UAVs by Broad Agency Announcement (BAA 06-06) in 2005 [3]. Furthermore, for PAVs, Wood et al. [4] defined them as UAVs with a size of within 5 cm and a mass of 500 mg. The performance of the small UAVs makes it possible for UAVs to enter spaces where people cannot enter and tight spaces in the case of a natural disaster, accident, or war, and to easily check the conditions of these areas using cameras and sensors. The aforementioned application potential is the development purpose of UAVs. In achieving this purpose, small UAVs are required not only to downsize but also to have high flight performance such as the ability to fly freely, high hovering power, and high flight stability to withstand wind in outdoor environments. Therefore, considerable research and development have been conducted to achieve downsizing and motion control [5,6,7]. These types of UAVs have been developed for research purposes: fixed-wing, rotary-wing, and flapping-wing. Among these, various studies on fixed-wing and rotary-wing UAVs have been conducted because they are similar to airplanes and helicopters and are easy to design. Fixed-wing UAVs can efficiently fly long distances; however, they are difficult to hover. On the contrary, rotary-wing UAVs have high hovering power and maneuverability, making them suitable for small UAV development [6]. However, fixed-wing UAVs cannot efficiently fly as the wing size is increased until they generate more lift [8], and rotary-wing UAVs do not show high flight performance because of their high power consumption [9] if they are downsized. On the contrary, flapping-wing UAVs can fly more efficiently than rotary-wing UAVs under low Reynolds number conditions [10], and these UAVs have been attracting attention recently. Flapping-wing UAVs were developed based on the flapping motion of birds and insects, which can not only generate enough lift to support their own weight but also have high flight performance, exhibited in such actions as making sharp turns and taking off swiftly in nature [11]. However, the development of flapping-wing UAVs remains in its nascent stages [6], and to advance the development of advanced small UAVs, deepening our understanding of the characteristics of the fluid surrounding insects and their flight is necessary, which have the same Reynolds number as small UAVs.

Insects have a high flight performance because of the high frequency and unsteady motion of their wings. However, the motion is completely different from the mechanism of lift generation by steady flow in airplanes and helicopters, and much has not been clarified because such a mechanism is difficult to explain using conventional aerodynamic principles. Therefore, considerable research on insect flight has been conducted to clarify the mechanism of lift generation in insects. Ellington et al. [12] indicated that a leading-edge vortex is a vortex that is generated from the leading edge of the wings, and more lift is generated by attaching this vortex to the wings. This factor primarily explains the mechanism of lift generation in insects, and it is common to all insects, including small and large insects [13,14]. In addition, regarding the flapping motion, Weis-Fogh [15] reported that clap and fling are the motions of opening and closing the wings, which generate more lift. Moreover, Cheng et al. [16] indicated that smaller insects perform a U-shaped upstroke motion, which generates more lift. Thus, we can understand the mechanism of lift generation in insects by investigating their flapping motion.

In the flapping motion of insects, simple motions such as line-shaped, oval-shaped, and the figure-eight trajectory of the wing tips (i.e., figure-eight motion) are often observed [17,18]. Considerable research on these motions has been conducted, and Aghav [19] and Lehmann et al. [20] indicated that figure-eight motion generated the most lift in these motions. Therefore, understanding the figure-eight motion will provide new knowledge for the development of UAVs that refer to the flapping motion of insects. Regarding the figure-8 motion, Galinski et al. [21] achieved an electromechanical device incorporating the figure-8 motion via experimentation. In addition, Ishihara [22] indicated that the figure-8 motion can be created via fluid–structure interaction (FSI) and that the lift increased significantly as the wing tip path mode shifted to the figure-eight mode. Moreover, many studies [20,23,24] have compared different strokes of insect wings, such as line-shaped, oval-shaped, figure-eight-shaped, and pear-shaped, and reported that the differences in their motions have a significant effect on the generation of lift. Therefore, we focused on the differences in the motion mode of the figure-eight motions in particular with the self-intersection of the wing-tip trajectories in detail. In addition, the motion is observed in insects of various sizes; however, it is especially observed in insects with low Reynolds numbers (Re≤130) [17,25]. Moreover, insects with high Reynolds numbers (Re≈720) move their wings horizontally (long and thin figure-eights) relative to the flapping plane, whereas insects with low Reynolds numbers move their wings vertically (thick figure-eights) with a large stroke deviation [17]. However, few studies have summarized the relationship between figure-eight motion and Reynolds number in detail. Therefore, this study aims to investigate the more efficient figure-eight motion patterns in generating lift during hovering motion related to the Reynolds number.

## 2. Materials and Methods

### 2.1. Model of an Insect

In this study, fruit fry was used as a model for analyzing the flapping of small insects. The parameters of the insect were based on the physical properties of fruit fry used by Aono et al. [26], and these values are shown in Table 1. The flight mode adopted hovering, which is the simplest flapping flight mode. In a previous study, Yamauchi et al. [27] reported that the body’s presence was not important in hovering because the presence has little effect on the cycle-averaged lift. Therefore, the analysis was performed with only two wings, without the reproduction of the insect body (Figure 1). Regarding the insect wing, it may be possible to perform a simplified analysis with only one wing using symmetric boundary conditions. However, in this study, it is necessary to capture the fluid dynamic interactions generated by each wing to consider the effects of the different motion modes of the wings. Therefore, the analysis was performed with two wings for higher accuracy of the results. The wings were reproduced with rectangular rigid plates without thickness to capture the shape of the wings on Cartesian grids as easily as possible. $S_{W}$ shown in Figure 1 is the wing area.

Moreover, the Reynolds number used in this study was defined as follows:(1)$$Re=\frac{\bar{u_{tip}}C_{m}}{\nu_{air}}.$$

Further, the analysis in Section 3.1 and Section 3.3 was performed with Re≈134 based on the values shown in Table 1. In addition, the mean velocity of the wing tip ($\bar{u_{tip}}=0.645$, 1.29, 2.58, 5.16, and 10.32) was varied and analyzed as Re≈33.5, 67, 134, 268, and 536 when the Reynolds number dependence was evaluated in Section 3.4. Note that the frequency of the wings (f=54.5, 109, 218, 436, and 872) was similarly varied when the mean velocity of the wing tip ($\bar{u_{tip}}=0.645$, 1.29, 2.58, 5.16, and 10.32) was varied.

### 2.2. Motion of the Insect

As shown in Figure 2, the flapping motion of an insect is represented by combining the rotational motion of the three axes. Each rotation angle is represented using Equations (2)–(4), where the positional angle, feathering angle, and elevation angle are around the z-axis, wing axis $x_{W}$, and y-axis, respectively. Note that the lapping motion of an insect was based on the study of fruit fry used by Aono et al. [26] as described in Section 2.1. In this study, a figure-eight motion was represented by varying the elevation motion angle $\theta_{e}$, which is a rotational motion that represents the figure-eight motion. The motions observed by the change of this angle were motions with the self-intersection of the wing-tip trajectory in the other motions ($0<\varphi_{e0}<180$ and $180<\varphi_{e0}$ < 360), except for the three line-shaped ($\theta_{e}=0{}^{\circ}$) and U-shaped ($\varphi_{e0}=0{}^{\circ}$ and $\varphi_{e0}=180{}^{\circ}$) motions. Therefore, in summarizing all the motions, we defined the motion with $\theta_{e}=0{}^{\circ}$ as a motion “without figure-eight motion” and the motion with $\theta_{e}\neq 0{}^{\circ}$ as a motion “with figure-eight motion”.
(2)$$\theta_{p}=\theta_{p,amp}\cos\left(2\pi ft\right)+\varphi_{p},$$
(3)$$\theta_{f}=\theta_{f,amp}\sin\left(2\pi ft\right)+\varphi_{f},$$
(4)$$\theta_{e}=\theta_{e,amp}\cos\left(2\pi \left(2f\right)t-\varphi_{e0}\right)+\varphi_{e},$$
where $\theta_{p}$ is the flapping angle, $\theta_{f}$ is the feathering angle, and $\theta_{e}$ is the elevation angle; $\theta_{p,amp}$, $\theta_{f,amp}$, and $\theta_{e,amp}$ are each amplitude; $\varphi_{p}$, $\varphi_{f}$, and $\varphi_{e}$ are each initial position; $\varphi_{e0}$ is the initial phase of the elevation angle and f is the flapping frequency. In this study, $\theta_{p,amp}$, $\theta_{f,amp}$, $\theta_{e,amp}$, $\varphi_{p}$, $\varphi_{f}$, $\varphi_{e}$, $\varphi_{e0}$, and f were set to 70°, 70°, 10°, 10°, 0°, 10°, 10°, and 218 Hz, respectively. Note that these parameters were also based on the properties of fruit fry used by Aono et al. [26] as described in Section 2.1. Based on the abovementioned conditions, Figure 3 shows the time history of the represented rotational motion angles $\theta_{p}$, $\theta_{f}$, and $\theta_{e}$, and the trajectories of the wing tips.

### 2.3. Governing Equation

In this study, the three-dimensional normalized lattice Boltzmann method [28,29] (3D27V model) with incompressible formulation was used as the governing equation for fluid resulting from three-dimensional flapping motion. The method was developed on the basis of the lattice Boltzmann method [30,31], which has the advantages of a simple algorithm and high computational efficiency. Thus, the proposed method can reduce memory usage and improve computational stability while maintaining the advantages of the lattice Boltzmann method.

The distribution function $f_{\alpha }$ in the lattice Boltzmann equation is expressed using a discrete velocity vector $e_{\alpha }$. In the normalized lattice Boltzmann equation, the incompressible Navier–Stokes equation can be represented using the second-order moments of the distribution function as follows:(5)$$f_{\alpha }\approx \omega_{\alpha }\left(a_{0}+b_{i}e_{\alpha i}+c_{ij}e_{\alpha i}e_{\alpha j}\right),$$
where $\omega_{\alpha }$ is the weight coefficient and $e_{\alpha }$ is the velocity vector, which are given in Table 2 for the advection direction α of the 3D27V model shown in Figure 4. Note that the 3D27V model was used in this study instead of the simple 3D19V model to capture more accurately complex flow fields in three dimensions.

In addition, $a_{0}$, $b_{i}$, and $c_{ij}$ in Equation (5) are constants that satisfy the following relations: (6)$$\rho =\sum_{\alpha }f_{\alpha },$$
(7)$$\rho u_{i}=\sum_{\alpha }e_{\alpha i}f_{\alpha },$$
(8)$$\Pi_{ij}^{neq}=\sum_{\alpha }e_{\alpha i}e_{\alpha j}f_{\alpha }-\frac{c^{2}}{3}\rho \delta_{ij}-\rho u_{i}u_{j},$$
where ρ is the fluid density, $\rho u_{i}$ is the moment, and $\Pi_{ij}^{neq}$ is the nonequilibrium part of the stress tensor. Thus, by substituting Equations (6)–(8) into Equation (5), the distribution function $f_{\alpha }$ can be expressed as follows:(9)$$f_{\alpha }=\omega_{\alpha }\rho \left[1+\frac{3\left(e_{\alpha i}u_{i}\right)}{c^{2}}+\frac{9{\left(e_{\alpha i}u_{i}\right)}^{2}}{2c^{4}}-\frac{3\left(u_{i}u_{i}\right)}{2c^{2}}\right]+\frac{9\omega_{\alpha }}{2c^{2}}\left(\frac{e_{\alpha i}e_{\alpha j}}{c^{2}}-\frac{1}{3}\delta_{ij}\right)\Pi_{ij}^{neq},$$
where the first term on the right is the equilibrium distribution function $f_{\alpha }^{eq}$ and the second term on the right is the nonequilibrium part of the distribution function $f_{\alpha }^{neq}$. Thus, they can be replaced as follows:(10)$$f_{\alpha }=f_{\alpha }^{eq}+f_{\alpha }^{neq},$$
(11)$$f_{\alpha }^{eq}=\omega_{\alpha }\rho \left[1+\frac{3\left(e_{\alpha i}u_{i}\right)}{c^{2}}+\frac{9{\left(e_{\alpha i}u_{i}\right)}^{2}}{2c^{4}}-\frac{3\left(u_{i}u_{i}\right)}{2c^{2}}\right],$$
(12)$$f_{\alpha }^{neq}=\frac{9\omega_{\alpha }}{2c^{2}}\left(\frac{e_{\alpha i}e_{\alpha j}}{c^{2}}-\frac{1}{3}\delta_{ij}\right)\Pi_{ij}^{neq}.$$

Further, the time evolution equation of the lattice Boltzmann equation can be expressed as follows:(13)$$f_{\alpha }\left(t+\delta t,x+e_{\alpha }\delta t\right)=f_{\alpha }\left(t,\;x\right)+\left(1-\frac{1}{\tau }\right)f_{\alpha }^{neq}\left(t,x\right),$$
where τ is the relaxation time, which is defined as follows:(14)$$\tau =\frac{3\nu }{c\delta x}+\frac{1}{2},$$
where ν is the kinematic viscosity.

As previously stated, the incompressibility formulation is applied to reduce the error caused by compressibility. In this case, the pressure distribution function $p_{\alpha }$ is defined using the following density distribution function $f_{\alpha }$:(15)$$p_{\alpha }=c_{s}^{2}f_{\alpha },$$
where $c_{s}$ is the speed of sound, which is defined as follows:(16)$$c_{s}=\frac{c}{\sqrt{3}}.$$

Thus, the pressure p, velocity component $u_{i}$, and nonequilibrium part of the stress tensor $\Pi_{ij}^{neq}$ can be expressed as follows:(17)$$p=\sum_{\alpha }p_{\alpha },$$
(18)$$u_{i}=\frac{1}{\rho c_{s}^{2}}\sum_{\alpha }e_{\alpha i}p_{\alpha },$$
(19)$$\Pi_{ij}^{neq}=\frac{1}{c_{s}^{2}}\left(\sum_{\alpha }e_{\alpha i}e_{\alpha j}p_{\alpha }-\frac{c^{2}}{3}\rho \delta_{ij}-\frac{c^{2}}{3}\rho u_{i}u_{j}\right).$$

Furthermore, the equilibrium distribution function $p_{\alpha }^{eq}$ can be expressed as follows:(20)$$p_{\alpha }^{eq}=\omega_{\alpha }\left[p+\rho_{0}\left\{\left(e_{\alpha }\cdot u\right)+\frac{3{\left(e_{\alpha }\cdot u\right)}^{2}}{2c^{2}}-\frac{u^{2}}{2}\right\}\right].$$

Thus, the time evolution equation of the lattice Boltzmann equation with the incompressibility formulation is defined as follows:(21)$$p_{\alpha }\left(t+\delta t,x+e_{\alpha }\delta t\right)=p_{\alpha }^{eq}\left(t,\;x\right)+\left(1-\frac{1}{\tau }\right)p_{\alpha }^{neq}\left(t,x\right).$$

### 2.4. Virtual Flux Method

In this study, the virtual flux method (VFM) [32,33,34] was used to represent insect wings on a Cartesian grid. This method has several advantages. It has a simple algorithm that is easy to implement, the physical quantities around the objects are accurately calculated, and the computational efficiency is higher than that of the immersed boundary method. As shown in Figure 5, the calculation of the VFM requires the placement of a virtual boundary point at the intersection of the boundary of the virtual object and the discrete velocity vector.

In this study, the no-slip condition for velocity and the Neumann boundary condition for pressure were used as the boundary conditions of the virtual object, and each boundary condition can be expressed as follows:(22)$$u_{vb}=u_{wall},$$
(23)$$\frac{\partial p}{\partial n}=0,$$
where $u_{wall}$ is the velocity of the object on the wall and n is the normal vector to the virtual boundary surface.

The calculation procedure for the VFM is shown in Figure 6. As shown in Figure 6, we consider the case in which an object at point E moves from point E to point D when calculating the distribution function for the next time step at point D. The object at point E cannot pass over the virtual boundary because the virtual boundary point is located at the point that divides the grid width into a :b between points D and E. Thus, we need to find the distribution function for the pressure at the next time step of point D using the distribution function at the virtual boundary point.

Pressure $p_{vb}$ at the virtual boundary can be calculated with extrapolation, and the numerical precision of the pressure is variable. In this study, we used extrapolation with second-order accuracy and pressure $p_{vb}$ was expressed by pressures $p_{1}$ and $p_{2}$ at points $h_{1}$ and $h_{2}$ away from the boundary point in the direction normal to the wall surface, which is presented as follows: (24)$$p_{vb}=\frac{h_{2}^{2}p_{1}-h_{1}^{2}p_{2}}{h_{2}^{2}-h_{1}^{2}},$$
where pressures $p_{1}$ and $p_{2}$ were interpolated with weights from the surrounding four grid points. In this case, $h_{1}$ and $h_{2}$ were set to $\sqrt{3}$ and $2\sqrt{3}$ times the grid width, respectively, to prevent the grid points from straddling the virtual boundary surface. The equilibrium distribution function $p_{\alpha }^{eq}$ was calculated by substituting the physical quantities $u_{vb}$ and $p_{vb}$ at the virtual boundary point into Equation (20) as follows: (25)$$p_{\alpha }^{eq}\left(t,x_{vb}\right)=\omega_{\alpha }\left[p_{vb}+\rho_{0}\left\{\left(e_{\alpha }\cdot \;u_{vb}\right)+\frac{3{\left(e_{\alpha }\cdot \;u_{vb}\right)}^{2}}{2c^{2}}-\frac{{\;u_{vb}}^{2}}{2}\right\}\right].$$

The virtual equilibrium distribution function $p_{\alpha }^{eq*}\left(t,x_{E}\right)$ at point E was linearly extrapolated from the equilibrium distribution function $p_{\alpha }^{eq}\left(t,x_{vb}\right)$, calculated in Equation (25), and the equilibrium distribution function $p_{\alpha }^{eq}\left(t,x_{D}\right)$ at point D from the internal ratio a:b shown in Figure 6. The nonequilibrium part of the virtual distribution function $p_{\alpha }^{neq*}\left(t,x_{E}\right)$ at point E was interpolated with the nonequilibrium part of the distribution function $p_{\alpha }^{neq}\left(t,x_{D}\right)$ at point D as follows: (26)$$p_{\alpha }^{eq*}\left(t,\;x_{E}\right)=\frac{a+b}{a}p_{\alpha }^{eq}\left(t,\;x_{vb}\right)-\frac{b}{a}p_{\alpha }^{eq}\left(t,\;x_{D}\right),$$
(27)$$p_{\alpha }^{neq*}\left(t,\;x_{E}\right)=p_{\alpha }^{neq}\left(t,\;x_{D}\right).$$

However, when the internal ratio a was small, the denominator was small and the calculation would possibly diverge. Therefore, in the case of a<0.5, the distribution function is calculated using the equilibrium distribution function and the nonequilibrium part of the distribution function at point C, which is the next point.

Thus, the distribution function $p_{\alpha }(t+\delta t,x_{D})$ at point D of the next time step can be calculated using the virtual equilibrium distribution function $p_{\alpha }^{eq*}\left(t,\;x_{E}\right)$ and the nonequilibrium part of the distribution function $p_{\alpha }^{neq*}\left(t,\;x_{E}\right)$ at point E as follows:(28)$$p_{\alpha }(t+\delta t,\;x_{D})=p_{\alpha }^{eq*}\left(t,\;x_{E}\right)+\left(1-\frac{1}{\tau }\right)p_{\alpha }^{neq*}\left(t,\;x_{E}\right).$$

### 2.5. Computational Model

Figure 7 shows a schematic view of the computational model used in this study. The computational domain has a size of 40L×40L×40L, where the representative length L is the mean chord length $C_{m}$ and the representative speed U is the mean velocity of the wing tip $\bar{u_{tip}}$. Note that a grid model consisting of four blocks of different grid sizes (four-tiered multiblock method [35]) was used to reduce computation time. In the case of the four-tiered block, each grid width was two times larger than the inner one and the coarsest grid width was eight times larger than the finest one. For the initial conditions of the computational domain, pressure p was set to 1/3 and velocity u was set to 0. For the boundary conditions of the computational domain (outside of block1), the pressure was fixed similarly to the initial condition in the x–z plane of y=40L, whereas the Neumann boundary condition was used in other planes for pressure and in all planes for velocity.

### 2.6. Evaluation Parameters

We calculated the lift coefficient $C_{L}$, thrust coefficient $C_{T}$, and power coefficient $C_{PWR}$ as the evaluation indices of the fluid in the flapping flight, as described in Section 2.1, to investigate the effects of the figure-eight motion on aerodynamics. They are defined as follows:(29)$$C_{L}=\frac{F_{z}}{\frac{1}{2}\rho_{air}{\bar{u_{tip}}}^{2}S_{W}},$$
(30)$$C_{T}=\frac{F_{y}}{\frac{1}{2}\rho_{air}{\bar{u_{tip}}}^{2}S_{W}},$$
(31)$$C_{PWR}=\frac{\sum \left(f_{local}\cdot u_{local}\right)}{\frac{1}{2}\rho_{air}{\bar{u_{tip}}}^{3}S_{W}},$$
where $F_{y}$ and $F_{z}$ are the forces in the y and z directions, $f_{local}$ is the force, and $u_{local}$ is the velocity at a certain point of the wing.

Moreover, we used the Q value, which expresses the vortex structure, and the helicity density $h_{d}$, which indicates the degree of twisting structure of the swirling flow by the vortex, as evaluation indices of the flow field. Such indices are defined as follows:(32)$$Q=\frac{1}{2}\left(\Omega_{ij}^{2}-S_{ij}^{2}\right),$$
(33)$$h_{d}=u\cdot \omega ,$$
where u is the velocity vector and ω is the vorticity vector. Moreover, $\Omega_{ij}$ is the vorticity tensor and $S_{ij}$ is the deformation velocity tensor, which are expressed as follows:(34)$$\Omega_{ij}=\frac{1}{2}\left(D_{ij}-D_{ji}\right)\;,$$
(35)$$S_{ij}=\frac{1}{2}\left(D_{ij}+D_{ji}\right)\;,$$
where D is the velocity gradient tensor, which is calculated as follows:(36)$$D=\left(\begin{array}{ccc} \frac{\partial u}{\partial x} & \frac{\partial u}{\partial y} & \frac{\partial u}{\partial z}\\ \frac{\partial v}{\partial x} & \frac{\partial v}{\partial y} & \frac{\partial v}{\partial z}\\ \frac{\partial w}{\partial x} & \frac{\partial w}{\partial y} & \frac{\partial w}{\partial z} \end{array}\right)=\left(\begin{array}{ccc} D_{11} & D_{12} & D_{13}\\ D_{21} & D_{22} & D_{23}\\ D_{31} & D_{32} & D_{33} \end{array}\right).$$

In this study, the Q value and helicity density $h_{d}$ are nondimensionalized as follows:(37)$$Q^{*}=\frac{Q}{{\left(U/L\right)}^{2}}\;,$$
(38)$$h_{d}^{*}=\frac{h_{d}}{\left|u\right|\cdot \left|\omega \right|}\;.$$

## 3. Results and Discussion

### 3.1. Grid Independence and Flapping Cycle Convergence Tests

A grid independence test was performed for three patterns with grid sizes of 16, 32, and 64 cells/L for the representative length of the finest grid size in the multiblock to investigate the number of grid sizes with numerical reliability in this analysis. The representative speed of the flapping flight U was set to 0.04, and the resolution was verified when the lift coefficient was stable in the seven flapping cycles of the flapping cycle convergence test. Figure 8 shows the time history of the lift coefficient in seven cycles, and Table 3 shows the cycle-averaged lift coefficient per cycle. As shown in Figure 8, a disturbance was observed at the beginning of the cycle because of the beginning of the wing movement; however, it converged immediately, and little change was found in the time history. In addition, not much difference was observed after the fifth cycle (Table 3). Therefore, the analysis conducted in this study was performed using the data from the sixth cycle.

For the grid independence test, Figure 9 shows the time history of the lift coefficient $C_{L}$, the pressure component of the lift coefficient $C_{L_{p}}$, and the viscous stress component the of lift coefficient $C_{L_{\tau }}$ over one cycle at each resolution, and Table 4 shows the cycle-averaged values of them $\bar{C_{L}}$, $\bar{C_{L_{p}}}$, and $\bar{C_{L_{\tau }}}$. Based on these results, the difference in lift at each resolution decreased as the resolution increased, and not much difference was observed between 32 and 64 cells/L. Therefore, a resolution of 32 cells/L was selected after considering the computational cost and accuracy.

### 3.2. Example Test for Analyzing the Three-Dimensional Flapping Motion

Fluid forces generated by a rectangular rigid plate in simple motion were compared with those in previous studies to verify the physical validity of the analysis results for a moving rigid plate in a three-dimensional analysis. The motion is oscillating, which is expressed through the following translational displacement (heaving motion) and rotation (flapping motion):(39)$$x(t)=L\sin\left(2\pi ft\right),$$
(40)$$\theta \left(t\right)=\frac{\pi }{2}-\frac{\pi }{4}\sin\left(2\pi ft+\frac{\pi }{3}\right),$$
where $x\left(t\right)$ is the displacement of the x-coordinate of the center of gravity and $\theta \left(t\right)$ is the oscillatory angle. Figure 10 shows a schematic view of the movement of an oscillating plate.

Figure 11 shows the time history of lift coefficient generated from an oscillatory plate. As shown in Figure 11, the simulated values were similar to the results of other studies [36,37]. In addition, the cycle-averaged value of the lift was 0.223 in this study, while the value was 0.22 in a previous study [36], with a difference of 1.3% and the results of this analysis were nearly consistent with the previous study.

### 3.3. Effect of Figure-Eight Motion

A comparison of the vortex structure and fluid forces generated by each motion was conducted to investigate the effect of the figure-eight motion on the aerodynamics. Each motion was represented by varying the elevation motion angle $\theta_{e}$ which is a rotational motion that represents the figure-eight motion shown in Equation (4). Motion with figure-eight motion was represented by the elevation motion angle with the parameters shown in Section 2.2, and motion without figure-eight motion was represented by $\theta_{e}=0$. Equations (41) and (42) show the angles with and without figure-eight motions as follows:(41)$$\theta_{e}=70\cos\left(2\pi \left(2f\right)t-10\right)+10,$$
(42)$$\theta_{e}=0.$$

Figure 12 shows a schematic view of each motion represented by the flapping motion angle $\theta_{p}$ and the feathering motion angle $\theta_{f}$ shown in Section 2.2 in addition to the elevation motion angle.

Figure 13 shows the isosurfaces of the vortex structure $Q^{*}=0.8$ formed via each motion at t=0.25T, 0.50T, 0.75T, and 1.00T. The isosurfaces were colored on the basis of the normalized helicity density $h_{d}^{*}$. As shown in Figure 13, the normalized helicity density was similar for each motion; however, the lengths of the vortex structures were different and the vortices with a figure-eight motion were longer than those without a figure-eight motion. This is because the vortices with a figure-eight motion stay longer on the top surface of the wings than those without a figure-eight motion, whereas the vortices without a figure-eight motion disappear immediately.

This difference in the vortex structure indicates a difference in the flow field, which indicates a difference in the relative velocity vector between the wings and the fluid. The difference in velocity vectors affects forces such as lift and thrust generated by the wings. Therefore, the time histories of the fluid forces generated via each motion were investigated. Figure 14 shows the time histories of the lift coefficient $C_{L}$, the thrust coefficient $C_{T}$, and the power coefficient $C_{PWR}$ over one stroke cycle for each motion, and Table 5 shows the cycle-averaged values of coefficients $\bar{C_{L}}$, $\bar{C_{T}}$, and $\bar{C_{PWR}}$ and the ratio of the lift coefficient to the power coefficient $\bar{C_{L}}/\bar{C_{PWR}}$ for each motion.

As shown in Figure 14a, a comparison between the lift coefficient with and without figure-eight motions showed a different trend throughout the entire cycle and an essentially different trend in the lift coefficient in the first half of each stroke. We examined the vortex structure and pressure coefficient distribution at t=0.1T, which corresponds to the initial stage of the downward motion of the wings (downstroke), to further discuss the cause of this large difference in the lift coefficient. Figure 15 shows the isosurfaces of the vortex structure ($Q^{*}=0.8$) formed by each motion. The isosurfaces were colored on the basis of the normalized pressure coefficient $C_{p}$. As shown in Figure 15, the motion with a figure-eight motion produced a larger leading-edge vortex and wing tip vortex on the upper surface of the wings than the motion without a figure-eight motion. This difference could be attributed to the fact that the z-directional motion of the figure-eight motion increased the angle relative to the motion direction (angle of attack) of the wings by adding a downward motion in the first half of the downstroke. Consequently, a larger negative pressure was produced on the upper surface of the wings, and the pressure difference between the positive pressure on the lower surface and the negative pressure on the upper surface of the wings generated a larger lift force than the motion without a figure-eight motion. In addition, as shown in Figure 14a, the relation of the lift coefficient in each motion reversed after the middle of the downstroke. This result was based on the abovementioned fact and was considered to be due to the fact that the figure-eight motion in the z-direction increases the angle of attack of the wings by adding upward motion after the middle of the downstroke. As shown in Table 5, the average lift coefficient with a figure-eight motion was approximately 26% larger than that without a figure-eight motion.

As shown in Figure 14b, a comparison between the thrust coefficient with and without figure-eight motions showed a similar trend throughout the entire cycle, and the cycle-averaged thrust coefficient in each was small (Table 5). Therefore, the effect of the figure-eight motion on the thrust coefficient was small. This effect could be attributed to the fact that the thrust coefficient was offset by the upward motion of the wings (upstroke) and downstroke.

As shown in Figure 14c, a comparison between the power coefficient with and without figure-eight motions showed a different trend throughout the entire cycle, and this difference could be attributed to the difference in the lift coefficient for each motion. As shown in Table 5, a comparison of the cycle-averaged power coefficient in each motion confirmed that the motion with figure-eight motion required more power. This finding could be attributed to the motion in the z-direction.

As shown in Table 5, the ratio of the lift coefficient to the power coefficient with a figure-eight motion was approximately 17% higher than that without a figure-eight motion. Therefore, these results indicated that a flight with a figure-eight motion consumed more power than a flight without a figure-eight motion but generated more lift, and it may be a more efficient way to fly while hovering.

### 3.4. Effect of Various Figure-Eight Motions and Reynolds Number

We compared the fluid forces generated via various figure-eight motions in addition to the motions shown in Section 3.3 to investigate the more efficient figure-eight motion patterns in generating lift during the hovering motion and the relationship between figure-eight motion and Reynolds number. These motions were represented as eight types of figure-eight motions by varying the initial phase $\varphi_{e0}$ of the elevation motion angle shown in Equation (4) from 0° to 315° in increments of 45°. The flapping motion angle and the feathering motion angle were the same as those shown in Section 2.2. Figure 16 shows a schematic view of each motion. Moreover, we investigated the dependence of the Reynolds number on the figure-eight motion.

Table 6 shows the cycle-averaged values of the lift coefficient and the ratio of the lift coefficient to the power coefficient for each motion at each Reynolds number. Figure 17 shows the relation between the cycle-averaged power coefficient and the cycle-averaged lift coefficient. Figure 17 shows the cycle-averaged power coefficient on the horizontal axis and the cycle-averaged lift coefficient on the vertical axis. Hence, comparing each type of motion, the motion that generates more lift for a certain amount of power (more efficient motion) is located in the upper left corner. This positional relationship through the comparison of motions can be described similarly by the slope of a line connecting the origin and one point of the results. Thus, the motion with a greater slope of the line is more efficient in generating lift. The dotted line shown in Figure 17 was used to compare each figure-eight motion and without a figure-eight motion, which was the line connecting the value of without-figure-eight motion and the origin of the graph. In this study, the line was known as “without-8-line”. The motion located to the upper left of “without-8-line” generated lift more efficiently than without a figure-eight motion, whereas the motion located to the lower right of “without-8-line” was less efficient than without a figure-eight motion.

As shown in Figure 16, the motions with $\varphi_{e0}=0{}^{\circ}$ and $\varphi_{e0}=180{}^{\circ}$ were recognized as U-shaped motions. However, as shown in Table 6, the most efficient motions were not the U-shaped motions but the figure-eight motions with $\varphi_{e0}=45{}^{\circ}$ at any Reynolds number. Therefore, in this study, the figure-eight motion was defined as the motion with $\theta_{e}\neq 0{}^{\circ}$, without discussing the U-shaped motion independently.

As shown in Table 6, the motion with a smaller initial phase of the elevation motion angle ($\varphi_{e0}\leq 90{}^{\circ}$) had a higher ratio of the lift coefficient to the power coefficient, and the motion could generate lift more efficiently. The motion with $\varphi_{e0}=90{}^{\circ}$ at Re=33.5 and $\varphi_{e0}=90{}^{\circ}$ at other Reynolds numbers was the most efficient in generating lift. As shown in Figure 16, this result could be attributed to the fact that the angle of attack was larger than the original figure-eight motion with $\varphi_{e0}=10{}^{\circ}$ by increasing the vertical motion of the wings in each stroke. On the contrary, the motion with $\varphi_{e0}=90{}^{\circ}$ had a larger vertical motion and a larger angle of attack compared with the motion with $\varphi_{e0}=45{}^{\circ}$; however, the upward motion was larger. Therefore, the motion was less efficient than the motion with $\varphi_{e0}=45{}^{\circ}$ when it was averaged over the entire stroke. However, horizontal forces dominate rather than vertical forces toward motion at low Reynolds numbers; therefore, increasing the vertical motion of the flapping motion and the angle of attack are important for generating more lift. Therefore, the motion with $\varphi_{e0}=90{}^{\circ}$ at Re=33.5 was the most efficient in generating lift. Moreover, the motion patterns of the most efficient in generating lift at each Reynolds number were investigated in detail. Figure 18 shows the relation between the initial phase of the elevation angle of the most efficient motion in generating lift and Reynolds number. Note that the most efficient motion in generating lift was calculated from the tangential point where $\bar{C_{L\;}}/\bar{C_{PWR}}$ was the greatest slope on the elliptical approximation shown in Figure 17. As shown in Figure 18, the initial phase of the elevation angle of the most efficient motion in generating lift increased as the Reynolds number decreased, whereas the angle decreased and approached zero as the Reynolds number increased.

In addition, as shown in Table 6 and Figure 17f, all motions generated more lift as the Reynolds number increased, and more lift was generated compared with the power. The above discussion is based on nondimensionalized values without considering differences in the speed of the wings (flapping frequency) to simplify the comparison of the lift and power generated by each motion. Then, the physical values of the lift and power coefficients were summarized to investigate the effect of the flapping frequency on aerodynamic characteristics. Table 7 shows the cycle-averaged values of the force in the z direction $F_{z}$, power P, and the ratio of the lift coefficient to the power coefficient $F_{z}/P$ generated via the motion without a figure-eight motion at each Reynolds number. As shown in Table 7, a higher flapping frequency generated more power than lift.

Moreover, as shown in Figure 17, the eight types of figure-eight motions showed an elliptical distribution in the relation between the lift coefficient and power coefficient. Therefore, we discuss the relation between the lift coefficient and power coefficient by making an elliptical approximation along the distribution. As shown in Figure 17a–e, the area surrounded by the ellipse and the W8 line (light blue shown in the figure) decreased as the Reynolds number increased, and the number of motions that generate lift more efficiently than without the figure-eight motion also decreased. This decrease indicates that the effect of figure-eight motion decreased as the Reynolds number increased. The elliptical shape, which is primarily related to the decrease in area, was focused on to discuss this difference in effect in more detail. Therefore, the effect of the change in the Reynolds number on the figure-eight motion was determined by calculating the aspect ratio and inclination angle of the ellipse at each Reynolds number. As shown in Figure 19, the long and short sides of the ellipse for each Reynolds number were defined as $l_{l}$ and $l_{s}$, respectively, and the aspect ratio AR can be expressed as $AR=l_{l}/l_{s}$, where AR is the ratio of the long side to the short side of the ellipse. In addition, the inclination angle of the ellipse and the location of the center of the ellipse in Figure 17 were defined as $\theta_{el}$ and O(x, y). Table 8 shows the values of the elliptic approximation, and Figure 20 shows the relation between the aspect ratio of the ellipse and the Reynolds number.

As shown in Table 8 and Figure 20, the inclination angle of the ellipse did not change remarkably; the aspect ratio increased with increasing Reynolds number, and the aspect ratio converged when the Reynolds number increased above a certain value. The increase in the aspect ratio indicated that the elliptical approximation of various figure-eight motions approached the straight “without-8-line” shown in Figure 17, which indicates that the difference between the motions with and without a figure-eight motion decreased. In addition, the increase in the Reynolds number was synonymous with the increase in insect size. Therefore, varying the elevation motion angle that represents the figure-eight motion was not very effective for insects under high Reynolds number conditions (large size). Particularly, the figure-eight motion may be a vital mechanism for insects to generate lift more efficiently under low Reynolds number conditions (small size).

In this study, insect wings were analyzed as rectangular rigid plates without thickness. Regarding the shape, Kirishna et al. [38] reported that flight efficiency did not change much depending on the shape of the wings between a rectangular model with a model based on the actual shape of a blowfly. However, the changes in the shape of the wings may affect the efficiency of each figure-eight motion pattern. Therefore, in future studies of insects, it is necessary to evaluate figure-eight motion by considering not only the shape of the wings but also the flexibility of the wings and other parameters.

## 4. Conclusions

In this study, we investigated the more efficient figure-eight motion patterns in generating lift during the hovering motion related to the Reynolds number. For this purpose, we compared the ratios of the cycle-averaged lift coefficient to the power coefficient generated via each motion by varying the elevation motion angle and Reynolds number. Consequently, the motion with a smaller initial phase of the elevation motion angle ($\varphi_{e0}\leq 90{}^{\circ}$) could generate lift more efficiently at all Reynolds numbers. In addition, the initial phase of the elevation angle of the most efficient motion in generating lift increased as the Reynolds number decreased, whereas the angle decreased and approached $\varphi_{e0}=0$ as the Reynolds number increased. Moreover, varying the elevation motion angle that represents the figure-eight motion was effective for insects under low Reynolds number conditions and was not very effective for insects under high Reynolds number conditions. Therefore, the figure-eight motion may be a vital mechanism for insects under low Reynolds number conditions to generate lift more efficiently.

## Figures and Tables

**Figure 1 biomimetics-09-00249-f001:**
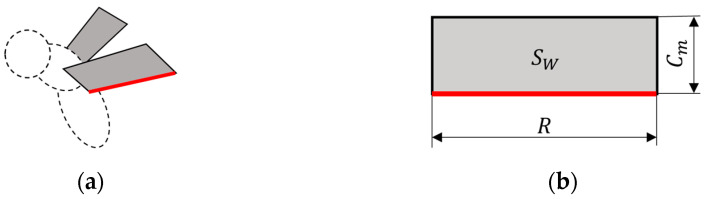
Computational model of an (**a**) insect and (**b**) wing. Only the two wings shown in gray were analyzed without body reproduction. The red line represents the wing length.

**Figure 2 biomimetics-09-00249-f002:**
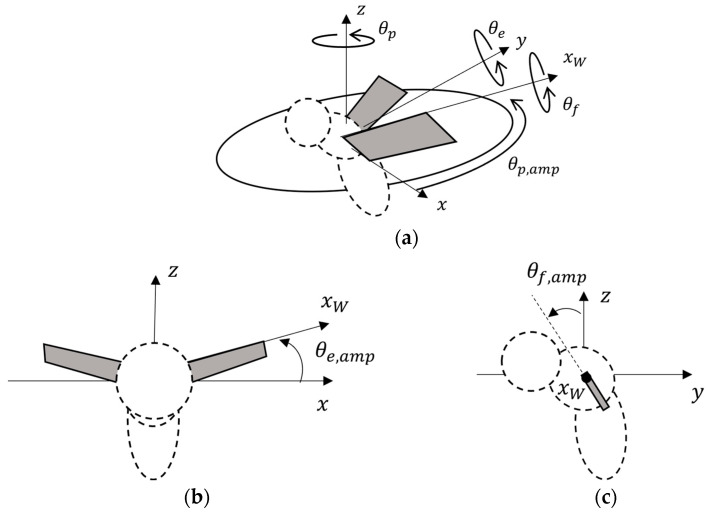
Definition of the flapping motion of the insect in (**a**) bird’s eye view, (**b**) x–z plane, and (**c**) y–z plane. The wings, which are the analysis objects, are shown in gray. On the other hand, the body, which is not the analysis object, is shown by the dotted line.

**Figure 3 biomimetics-09-00249-f003:**
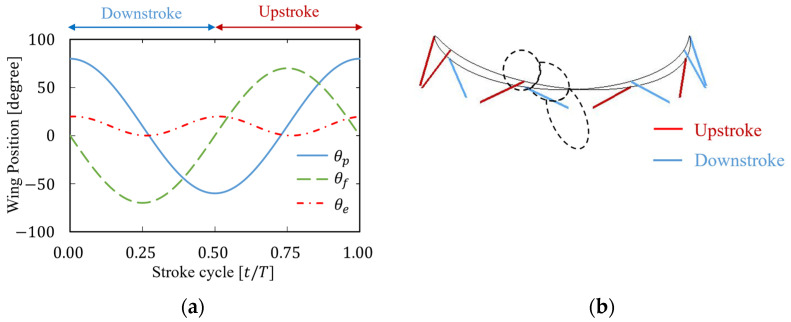
Representation of insect flapping motion. (**a**) Time history of the positional angle, feathering angle, and elevation angle. (**b**) Trajectory of wing tips. In Figure 3**b**,The gray solid and dotted lines represent the trajectory of the wings tips and the body of the insect, respectively.

**Figure 4 biomimetics-09-00249-f004:**
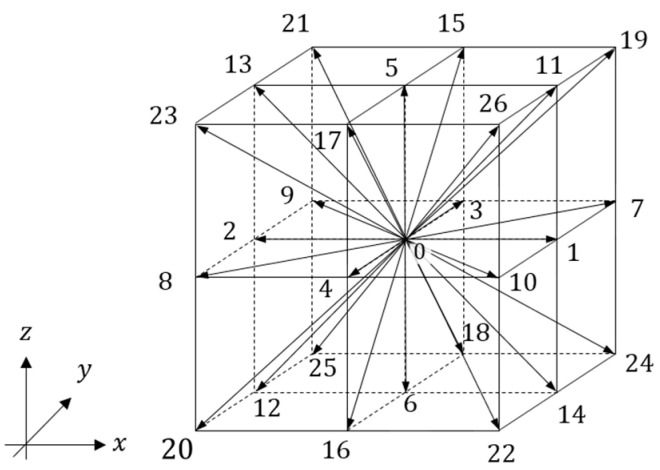
3D27V model for the three-dimensional lattice Boltzmann method. The numbers (0~26) represent the 27 directions of the 3D27Vmodel.

**Figure 5 biomimetics-09-00249-f005:**
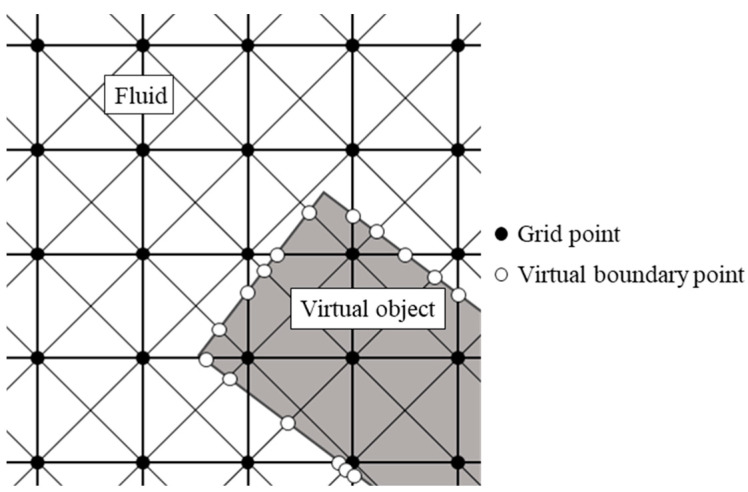
Virtual boundary points. The gray and white areas represent the interior of the object and fluid, respectively.

**Figure 6 biomimetics-09-00249-f006:**
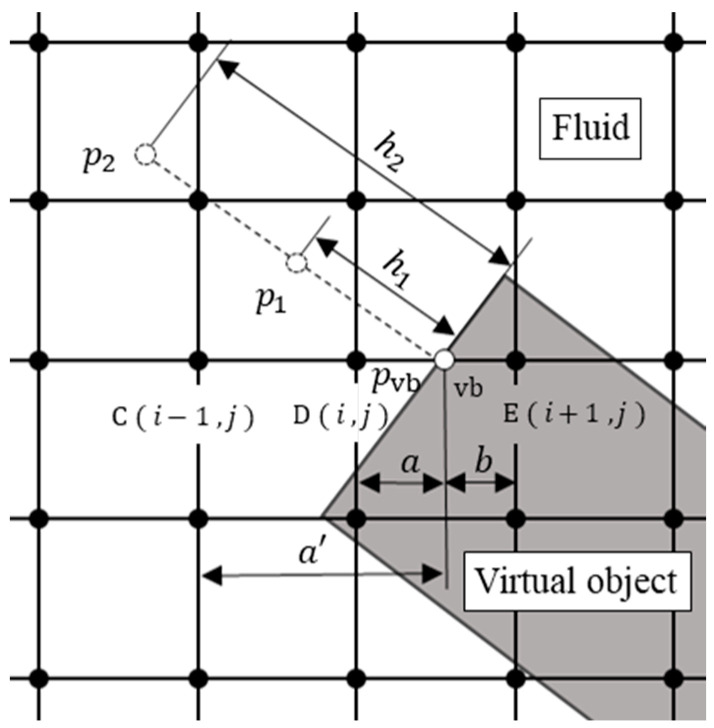
Schematic view of the physical quantity calculation method at the virtual boundary point. The gray and white areas represent the interior of the object and fluid, respectively.

**Figure 7 biomimetics-09-00249-f007:**
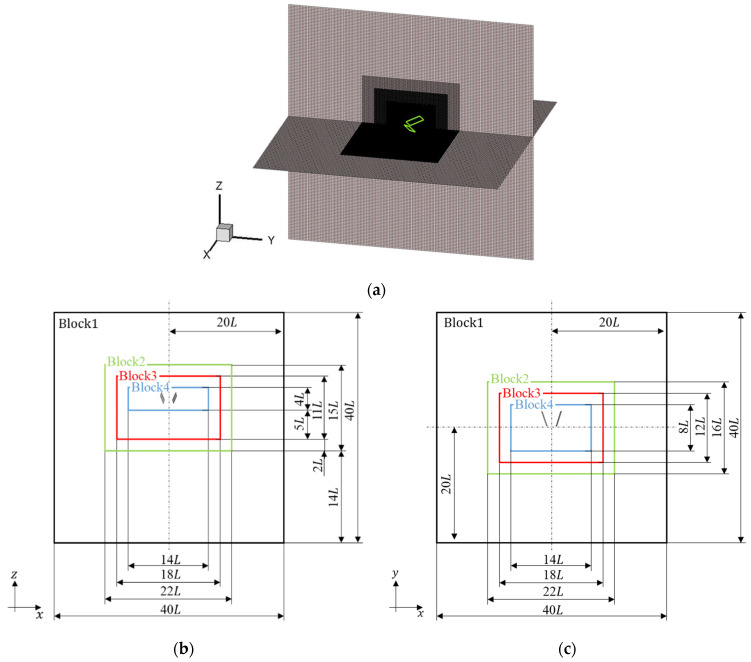
Schematic view of the computational model: (**a**) bird’s eye view, (**b**) x–z plane, and (**c**) x–y plane. The representative length L is the mean chord length $C_{m}$ shown in Section 2.1. In Figure 7**a**, the wings and grid model are shown in green and black or gray lines, respectively. In Figure 7**b**,**c**, the wings are shown in gray areas, and block areas of grid model consisting of different grid sizes are represented by black, green, red, and blue lines.

**Figure 8 biomimetics-09-00249-f008:**
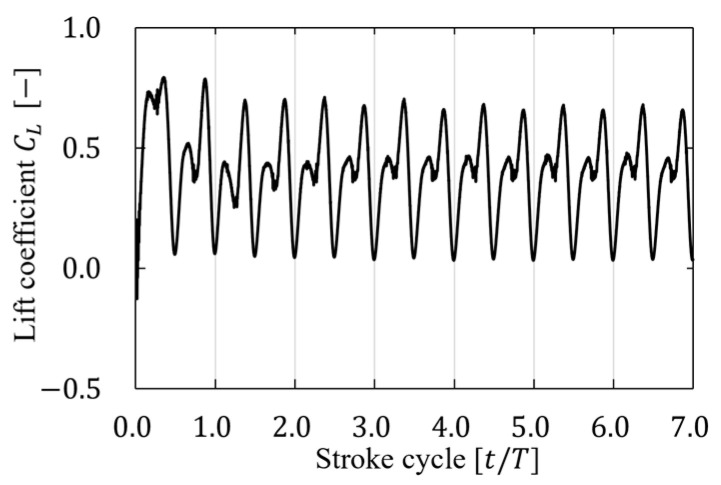
Time history of the lift coefficient in seven cycles with U=0.04.

**Figure 9 biomimetics-09-00249-f009:**
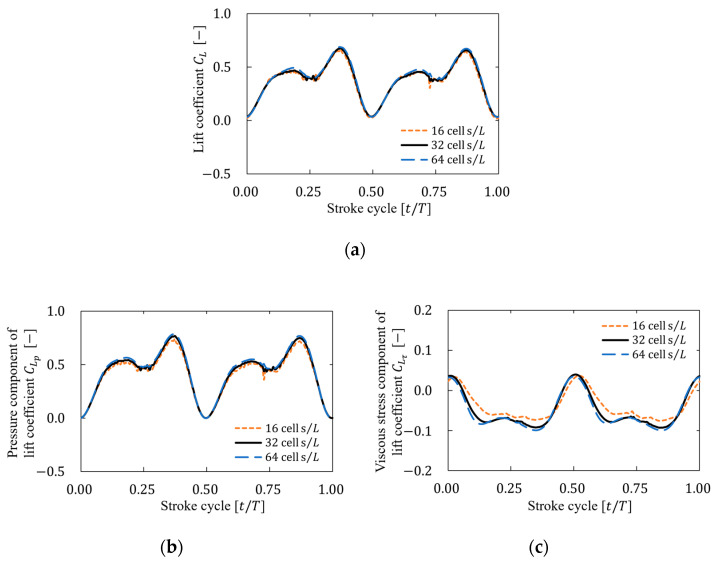
Time history of the (**a**) lift coefficient, (**b**) pressure component of lift coefficient, and (**c**) viscous stress component of lift coefficient for three resolutions.

**Figure 10 biomimetics-09-00249-f010:**
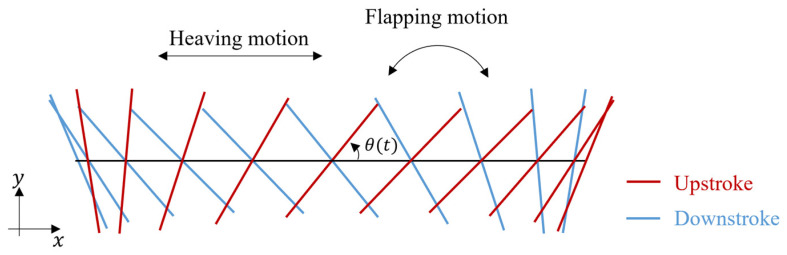
Schematic view of the movement of an oscillating plate.

**Figure 11 biomimetics-09-00249-f011:**
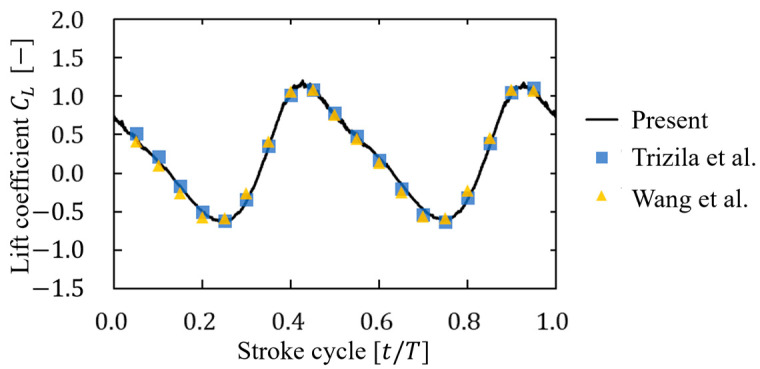
Time history of the lift coefficient of an oscillating plate. The result was compared with those of Trizila [36] and Wang et al. [37].

**Figure 12 biomimetics-09-00249-f012:**
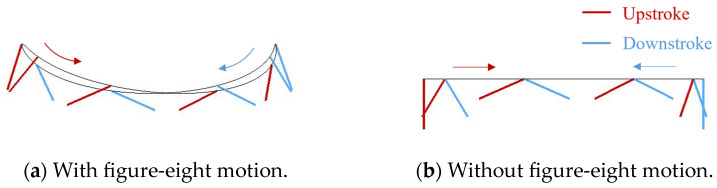
Trajectory of the wing tip in each motion. (**a**) With figure-eight motion and (**b**) without figure-eight motion. The gray line represents the trajectory of the wing tip.

**Figure 13 biomimetics-09-00249-f013:**
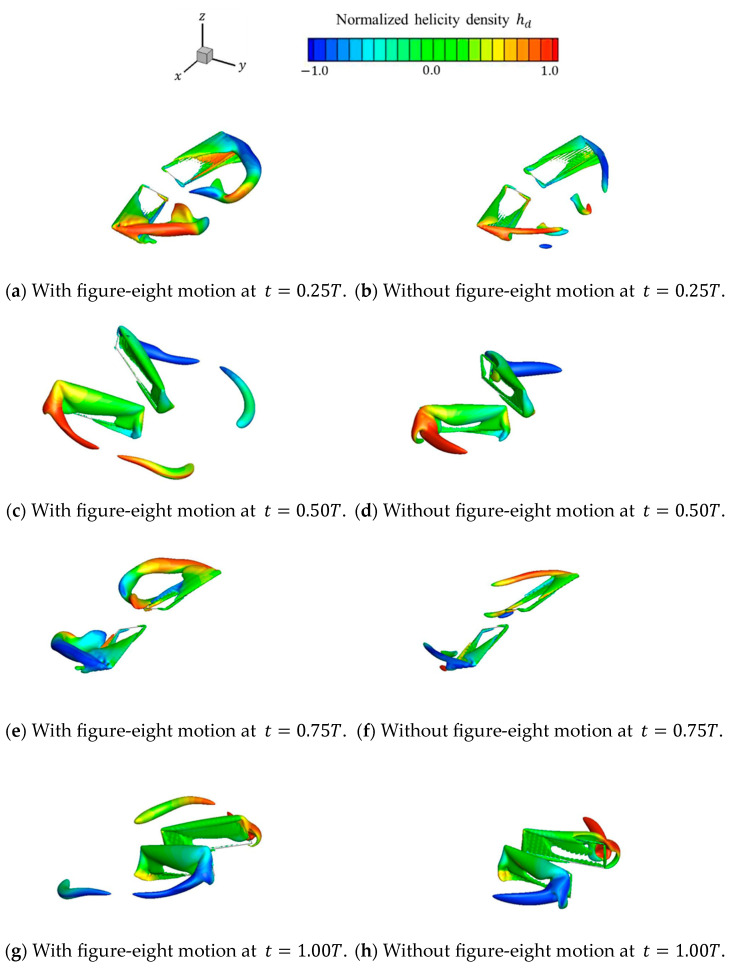
Vortex structures and normalized helicity density over one stroke cycle for each motion. (**a**) With and (**b**) without figure-eight motions at t=0.25T; (**c**) with and (**d**) without figure-eight motions at t=0.50T; (**e**) with and (**f**) without figure-eight motions at t=0.75T; and (**g**) with and (**h**) without figure-eight motions at t=1.00T.

**Figure 14 biomimetics-09-00249-f014:**
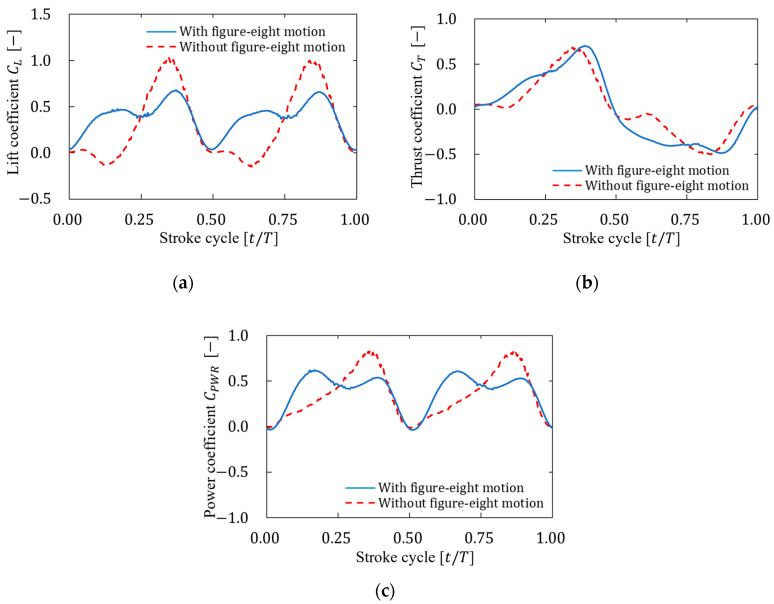
Time histories of (**a**) lift coefficient, (**b**) thrust coefficient, and (**c**) power coefficient over one stroke cycle for each motion.

**Figure 15 biomimetics-09-00249-f015:**
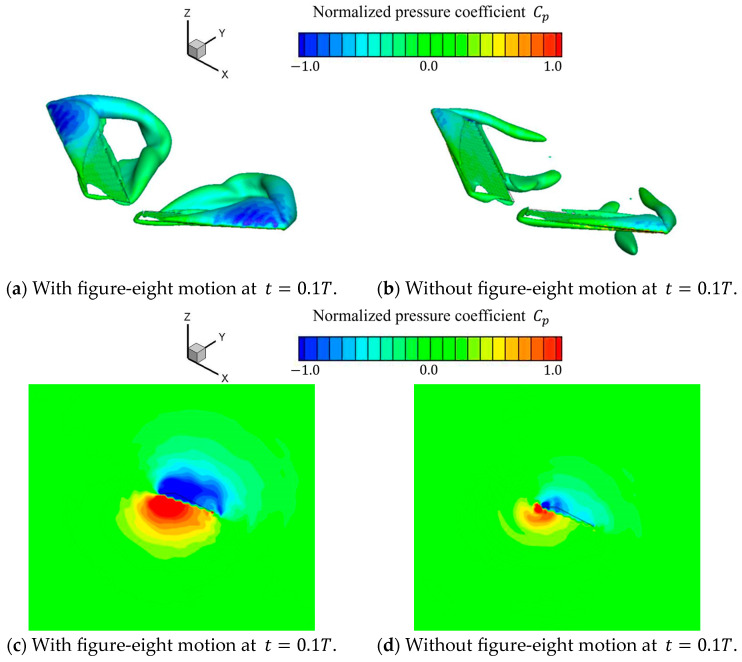
Vortex structures ($Q^{*}=0.8$) and normalized pressure coefficient for each motion in the downstroke at t=0.1T. Vortex structures ($Q^{*}=0.8$) and normalized pressure coefficients from (**a**) with and (**b**) without figure-eight motions. Normalized pressure coefficient at the wing tip from (**c**) with and (**d**) without figure-eight motions.

**Figure 16 biomimetics-09-00249-f016:**
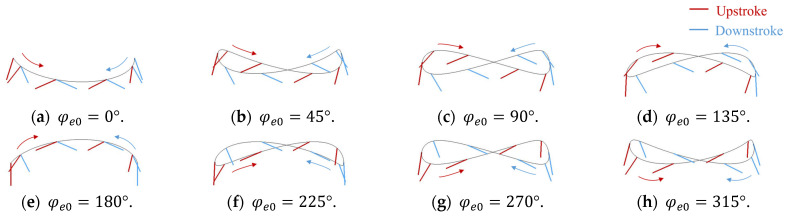
Trajectory of the wing tip in each motion: (**a**) $\varphi_{e0}=0{}^{\circ}$, (**b**) $\varphi_{e0}=45{}^{\circ}$; (**c**) $\varphi_{e0}=90{}^{\circ}$; (**d**) $\varphi_{e0}=135{}^{\circ}$; (**e**) $\varphi_{e0}=180{}^{\circ}$; (**f**) $\varphi_{e0}=225{}^{\circ}$; (**g**) $\varphi_{e0}=270{}^{\circ}$; and (**h**) $\varphi_{e0}=315{}^{\circ}$. The gray line represents the trajectory of the wing tip.

**Figure 17 biomimetics-09-00249-f017:**
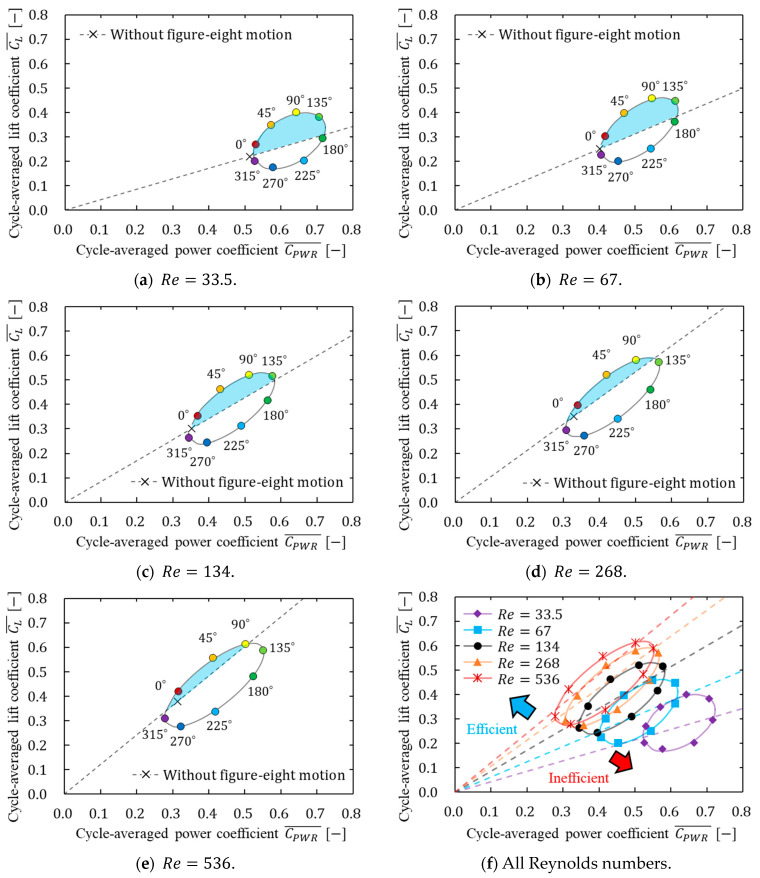
Relation between the cycle-averaged lift coefficient and power coefficient for each motion at each Reynolds number. (**a**) Re=33.5; (**b**) Re=67; (**c**) Re=134; (**d**) Re=268; (**e**) Re=546; (**f**) all Reynolds numbers. Light blue areas shown in the figure (
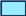
) indicate more efficiency than without figure-eight motions.

**Figure 18 biomimetics-09-00249-f018:**
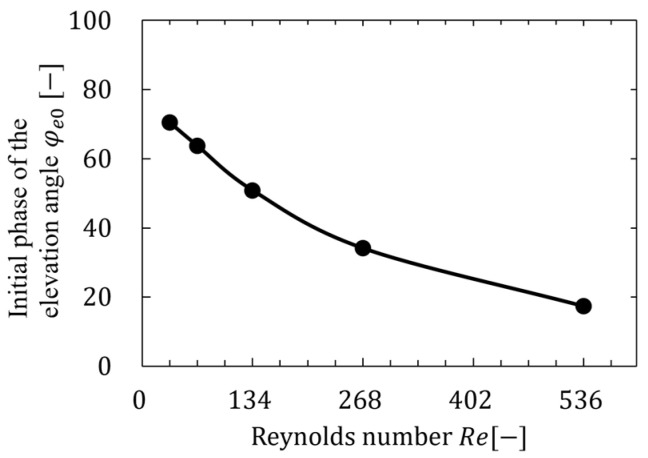
Relation between the initial phase of elevation angle of the most efficient motion in generating lift and Reynolds number.

**Figure 19 biomimetics-09-00249-f019:**
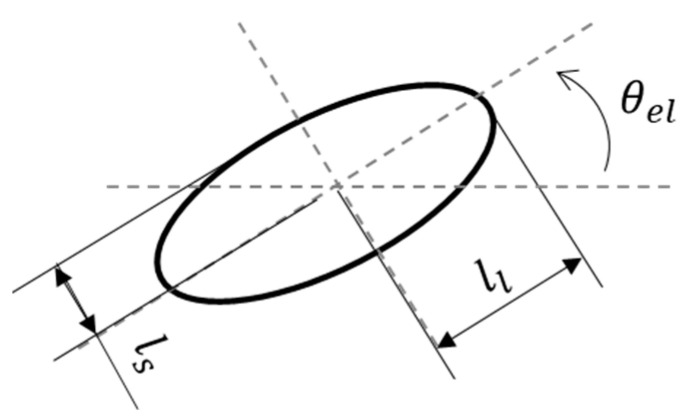
Schematic view of the elliptic approximation.

**Figure 20 biomimetics-09-00249-f020:**
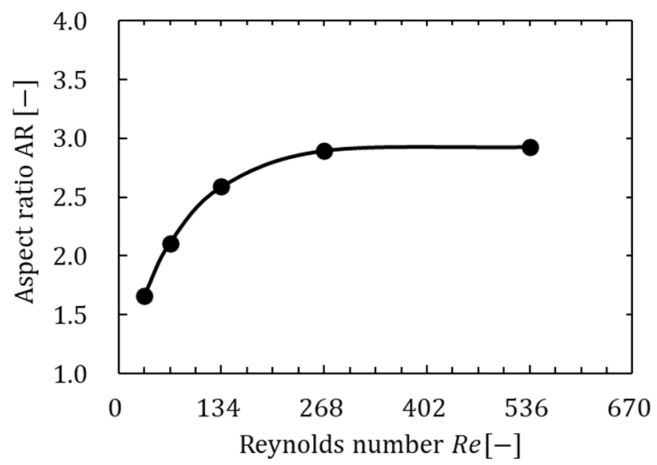
Relation between the aspect ratio and Reynolds number.

**Table 1 biomimetics-09-00249-t001:** Physical properties of an insect.

Name	Symbol	Value
Body length	$l_{b}$	2.78 [mm]
Mean chord length	$C_{m}$	0.78 [mm]
Wing length	R	2.39 [mm]
Mean velocity of wing tip	$\bar{u_{tip}}$	2.58 [m/s]
Wingbeat frequency	f	218 [Hz]
Kinematic viscosity of air	$\nu_{air}$	$1.5\times 10^{-5}\;[m^{2}/s]$
Density of air	$\rho_{air}$	$1.2\;[kg/m^{3}]$

**Table 2 biomimetics-09-00249-t002:** Advection parameters of the particles.

α	$e_{\alpha }$	$\left|e_{\alpha }\right|$	$\omega_{\alpha }$
0	$\left(0,0,0\right)$	0	8/27
1–6	$\left(\pm 1,0,0\right)$, $\left(0,\pm 1,0\right)$, $\left(0,0,\pm 1\right)$	c	2/27
7–18	$\left(\pm 1,\pm 1,0\right)$, $\left(\pm 1,0,\pm 1\right)$, $\left(0,\pm 1,\pm 1\right)$	$\sqrt{2}c$	1/54
19–26	$\left(\pm 1,\pm 1,\pm 1\right)$	$\sqrt{3}c$	1/216

**Table 3 biomimetics-09-00249-t003:** Cycle-averaged lift coefficient in seven cycles.

Cycle	$\bar{C_{L}}$
1	0.484
2	0.379
3	0.390
4	0.387
5	0.383
6	0.382
7	0.383

**Table 4 biomimetics-09-00249-t004:** Cycle-averaged lift coefficient, pressure component of lift coefficient, and viscous stress component of lift coefficient for three resolutions.

Resolution	$\bar{C_{L}}$	$\bar{C_{L_{p}}}$	$\bar{C_{L_{\tau }}}$
16 cells/L	0.373	0.408	0.038
32 cells/L	0.382	0.427	0.048
64 cells/L	0.392	0.442	0.052

**Table 5 biomimetics-09-00249-t005:** Cycle-averaged values of lift coefficient, thrust coefficient, power coefficient, and the ratio of the lift coefficient to the power coefficient. (A) With figure-eight motion and (B) without figure-eight motion.

Case	$\bar{C_{L}}$	$\bar{C_{T}}$	$\bar{C_{PWR}}$	$\bar{C_{L}}/\bar{C_{PWR}}$
(A)	0.382	$3.45\times 10^{-3}$	0.380	1.005
(B)	0.302	$2.54\times 10^{-2}$	0.352	0.857

**Table 6 biomimetics-09-00249-t006:** Cycle-averaged values of lift coefficient and the ratio of the lift coefficient to the power coefficient for each motion at each Reynolds number. (A) With figure-eight motion; (B) without figure-eight motion. Cycle-averaged values with a figure-eight motion (
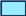
) indicate more efficiency than those without figure-eight motions.

		CL¯	CL¯/CPWR¯
Case		*Re*	33.5	67	134	268	536	33.5	67	134	268	536
φe0	
(A)	0	0.270	0.303	0.354	0.396	0.421	0.509	0.726	0.961	1.165	1.335
45	0.350	0.398	0.463	0.522	0.558	0.613	0.849	1.075	1.246	1.358
90	0.401	0.458	0.520	0.581	0.613	0.624	0.838	1.019	1.159	1.223
135	0.383	0.448	0.516	0.572	0.589	0.542	0.733	0.895	1.014	1.069
180	0.296	0.363	0.417	0.460	0.483	0.413	0.594	0.741	0.851	0.922
225	0.203	0.252	0.312	0.341	0.336	0.306	0.463	0.637	0.757	0.803
270	0.177	0.202	0.246	0.274	0.278	0.306	0.447	0.623	0.766	0.864
315	0.202	0.226	0.265	0.294	0.310	0.384	0.558	0.768	0.956	1.116
(B)	-	0.220	0.251	0.302	0.350	0.379	0.429	0.623	0.857	1.065	1.207

**Table 7 biomimetics-09-00249-t007:** Cycle-averaged values of the force in the z direction $F_{z}$, power P, and the ratio of the force in the z direction to the power $F_{z}/P$ without figure-eight motion at each Reynolds number.

Re [−]	$F_{z}$ [μN]	P [μW]	$F_{z}/P$ [−]
33.5	0.205	0.308	0.665
67	0.933	1.932	0.483
134	4.495	13.53	0.332
268	20.84	101.0	0.206
536	90.32	772.3	0.117

**Table 8 biomimetics-09-00249-t008:** Respective parameters of the elliptic approximation for each Reynolds number.

Re	O(x,y)	$l_{l}$	$l_{s}$	$\theta_{el}$	AR
33.5	(0.623, 0.284)	0.131	0.079	54	1.658
67	(0.511, 0.329)	0.154	0.073	54	2.110
134	(0.468, 0.385)	0.176	0.068	53	2.588
268	(0.437, 0.429)	0.197	0.068	53	2.897
536	(0.415, 0.446)	0.208	0.071	53	2.930

## Data Availability

Dataset available on request from the authors.

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
