# Peer review of "Numerical Simulation of the Advantages of the Figure-Eight Flapping Motion of an Insect on Aerodynamics under Low Reynolds Number Conditions"

_biomimetics, 2024, doi:10.3390/biomimetics9040249_

Round 1

Reviewer 1 Report

Comments and Suggestions for Authors

Please see the below file

Comments on the Quality of English Language

Minor editing of English language required

Reviewer 2 Report

Comments and Suggestions for Authors

The manuscript compares the aerodynamic performance of flapping wings that use follow different kinematic protocols. I recommend publication if the authors address the following comments:

1. The authors refer to the figure-eight flapping motion but, in fact, what seems to matter is the elevation motion but not the self-intersection of the wing-tip trajectories. For example, pure U-shape trajectories are more efficient than the planar trajectories and they are good for energy saving.

2. The nominally “figure-8” considered in section 3.3 is also close to the pure U-shape. I suggest that the authors make a U-shape trajectory out of the contour shown in Fig. 12a by averaging the upstroke and the downstroke, and report the changes in the aerodynamic force.

3. For the above, I suggest that refer to the non-zero elevation or stroke plane deviation rather than the figure-8 path. Especially in the title.

4. Placement of the pivot line at the leading edge is unnatural. The authors should motivate this choice.

5. Section title “3.4. Effect of various and Reynolds number” Various what?

6. Figure 18, table 7: the location of the center of the ellipse is also necessary to fully characterize the contours, these data must be included in the table.

7. Line 88: Liu et al. [19] -> Aono et al. [19]

8. Section 3.4: similar types of wing-tip trajectories with and without clap-and-fling were considered by Lehmann and Pick  dx.doi.org://10.1242/jeb.02746  I suggest that the authors comment on the similarities and differences.

Comments on the Quality of English Language

None

Reviewer 3 Report

Comments and Suggestions for Authors

  1. How does the lift coefficient-to-power coefficient ratio vary with changes in the elevation motion angle at different Reynolds numbers in the figure-eight flapping motion of insect wings?
  2. What are the underlying fluid dynamics principles that contribute to the increased lift efficiency observed in the figure-eight flapping motion of insect wings under low Reynolds number conditions?
  3. Can computational fluid dynamics simulations accurately capture the intricate aerodynamic effects of the figure-eight motion of insect wings, especially at low Reynolds numbers?
  4. How do the findings of this study on the advantages of figure-eight motion in insect flight contribute to the design and optimization of micro air vehicles (MAVs) operating at low Reynolds numbers?
  5. Figure 4 requires more details in the manuscript.
  6. Are there any biomechanical constraints or limitations that may hinder the implementation of figure-eight motion in artificial flapping-wing systems for MAVs, considering the observed trade-offs between lift generation and power consumption?
  7. Figure 11 Add difference % Or Error Bar to show the difference between the present study and literature.
  8. What are the implications of the figure-eight motion being more effective at lower Reynolds numbers for the evolutionary adaptation and ecological niche occupation of insect species?
  9. How do variations in wing morphology and kinematics among different insect species influence the efficacy of figure-eight motion in generating lift under varying Reynolds number conditions?
  10. Can the observed trends in lift generation and power consumption in this study be generalized across a broader range of insect species, or are there specific adaptations unique to certain taxa that affect the aerodynamic performance?
  11. What are the potential applications of the insights gained from studying the aerodynamic advantages of figure-eight motion in insect flight beyond the field of micro air vehicles, such as in biomimetic design, wind energy harvesting, or environmental monitoring systems?
  12. How do external factors such as turbulence, unsteady flow conditions, or environmental disturbances affect the performance of insects utilizing figure-eight motion for flight, and how do they adapt to these challenges?
Comments on the Quality of English Language

Minor editing of the English language required

Reviewer 4 Report

Comments and Suggestions for Authors

This research simulates the aerodynamics of a flapping plate under figure-eight wingtip trajectories. The results showed a higher lift efficiency at a figure-eight trajectory and this profit is more prominent at a low Reynolds number. Although the authors conclude that previous relevant works are based on two-dimensional analysis, there still exists some three-dimensional research about the deviation/elevation of an insect wing. Thus, I do not agree with the main motivation of this work and recommend a resubmission after consideration of the following works,

1.      Jung, Hyunwoo, Sehyeong Oh, and Haecheon Choi. "Role of the deviation motion on the aerodynamic performance of a mosquito wing in hover." Computers & Fluids 270 (2024): 106146.

2.      Hu, Fujia, et al. "Effects of asymmetric stroke deviation on the aerodynamic performance of flapping wing." Proceedings of the Institution of Mechanical Engineers, Part G: Journal of Aerospace Engineering 237.2 (2023): 480-499.

3.      Yao, J., and K. S. Yeo. "Free hovering of hummingbird hawkmoth and effects of wing mass and wing elevation." Computers & Fluids 186 (2019): 99-127.

4.      Shahzad, Aamer, et al. "Effects of wing shape, aspect ratio and deviation angle on aerodynamic performance of flapping wings in hover." Physics of Fluids 28 (2016): 111901.

5.      Luo, Guoyu, Gang Du, and Mao Sun. "Effects of stroke deviation on aerodynamic force production of a flapping wing." AIAA Journal 56.1 (2018): 25-35.

6.      Kim, Ho-Young, Jong-Seob Han, and Jae-Hung Han. "Aerodynamic effects of deviating motion of flapping wings in hovering flight." Bioinspiration & Biomimetics 14.2 (2019): 026006.

7.      Hu, Fujia, and Xiaomin Liu. "Effects of stroke deviation on hovering aerodynamic performance of flapping wings." Physics of Fluids 31. (2019): 111901.

Comments on the Quality of English Language

Notihing

Round 2

Reviewer 1 Report

Comments and Suggestions for Authors

Dear Authors,

I thank the authors for responding to my comments. However, I am of the opinion that some of them have not been adequately addressed. Please consider the following points:

#1. Literature survey about the figure-8 motions still sounds too coarse to reach the aim of this study. Especially, the authors need to address the numerical studies more carefully, since this study is a pure numerical one. In the revision, However, in many previous studies, comparisons of various motions with stoke deviation have been conducted [20-22], few studies have focused on in particular the figure eight motion patterns in detail. This line is quite incorrect. The meaning of various motions with stoke deviation is quite unclear, and there exist the studies that addressed the figure-8 motion patterns in detail as pointed out in my previous comment.

#5. I requested a discussion on the reason why a pair of wings is used instead of using one wing. The question on this point will be given by potential readers naturally because the numerical model assumed the ignorable body existence that leads to the sufficiency of only one wing. The discussion on this point will be useful for the readers who study the related topics.

Sincerely,

Reviewer

Comments on the Quality of English Language

Minor editing of English language required

Reviewer 3 Report

Comments and Suggestions for Authors

The authors have incorporated the comments into the revised version of the manuscript.
